# Alpha-Cypermethrin Resistance in *Musca domestica*: Resistance Instability, Realized Heritability, Risk Assessment, and Insecticide Cross-Resistance

**DOI:** 10.3390/insects14030233

**Published:** 2023-02-26

**Authors:** Naeem Abbas, Abdulwahab M. Hafez

**Affiliations:** Pesticides and Environmental Toxicology Laboratory, Department of Plant Protection, College of Food and Agriculture Sciences, King Saud University, Riyadh 11451, Saudi Arabia

**Keywords:** public health, house fly, vector-borne disease, integrated vector management

## Abstract

**Simple Summary:**

The common house fly, *Musca domestica* L., is a major carrier of serious diseases in humans and livestock. The common house fly has developed resistance to many insecticides used against it. In the present study, resistance to alpha-cypermethrin increased from 46.4-fold to 474.2-fold in alpha-cypermethrin-selected (Alpha-Sel) females and 41.0-fold to 253.2-fold in Alpha-Sel males, when compared with an alpha-cypermethrin-unselected strain (Alpha-Unsel). However, alpha-cypermethrin resistance was unstable when a field population was reared without exposure for 24 generations. The realized heritability (*h*^2^) of alpha-cypermethrin resistance was 0.17 and 0.18 for males and females, respectively, in G_1_–G_24_. The Alpha-Sel strain revealed low cross-resistance (CR) to two pyrethroids and five organophosphates and moderate CR to bifenthrin (15.5-fold), deltamethrin (28.4-fold), or cyfluthrin (16.8-fold). The results of instability of resistance trait, low *h*^2^, and lack of CR associated with alpha-cypermethrin resistance will provide an opportunity to stakeholders and entomologists to plan better and more effective insect pest and vector management programs in Saudi Arabia.

**Abstract:**

*Musca domestica* L., the common house fly, is a cosmopolitan carrier of human and livestock disease pathogens. The species exhibits resistance to many insecticides; therefore, effective *M. domestica* insecticide resistance management programs are required worldwide. In the present study, the development of alpha-cypermethrin resistance, realized heritability (*h*^2^), instability of resistance trait (DR), and cross-resistance (CR) was investigated in an alpha-cypermethrin-selected *M. domestica* strain (Alpha-Sel) across 24 generations (Gs). Compared with an alpha-cypermethrin-unselected strain (Alpha-Unsel), resistance to alpha-cypermethrin increased from 46.4-fold (G_5_) to 474.2-fold (G_24_) in Alpha-Sel females and 41.0-fold (G_5_) to 253.2-fold (G_24_) in Alpha-Sel males. Alpha-cypermethrin resistance declined by between –0.10 (G_5_) and –0.05 (G_24_) in both *M. domestica* sexes without insecticide exposure for 24 generations. The *h*^2^ of alpha-cypermethrin resistance was 0.17 and 0.18 for males and females, respectively, in G_1_–G_24_. With selection intensities of 10–90%, the G values required for a tenfold increase in the LC_50_ of alpha-cypermethrin were 6.3–53.7, 4.1–33.8, and 3.0–24.7, given *h*^2^ values of 0.17, 0.27, and 0.37, respectively, and a constant slope of 2.1 for males and *h*^2^ values of 0.18, 0.28, and 0.38, respectively, and a constant slope of 2.0 for females. Compared with Alpha-Unsel, Alpha-Sel *M. domestica* exhibited moderate CR to bifenthrin (15.5-fold), deltamethrin (28.4-fold), and cyfluthrin (16.8-fold), low CR to two pyrethroids and five organophosphates, and no CR to insect growth regulators. The instability of resistance trait, low *h*^2^, and absent or low CR associated with alpha-cypermethrin resistance in *M. domestica* indicate resistance could be managed with rotational use of the insecticide.

## 1. Introduction

Synthetic pyrethroid insecticides are commonly used to manage vector pests worldwide owing to their efficacy against the pests’ adult and larval stages, lack of persistence in the environment, and low mammalian toxicity [1,2]. However, the indiscriminate use of and over-reliance on these insecticides can increase environmental pollution, negatively affect human health through exposure, and lead to insecticide resistance in the target insect vectors [3,4,5,6,7]. Following the development of insecticide resistance, the public may increase the dosages of insecticides used to suppress resistant insect vectors, thereby compounding the negative effects on the environment and fauna [8,9]. Therefore, characterization of insecticide resistance development is necessary to manage resistant insect vectors and minimize insecticide-related effects on the environment and nontarget organisms [10].

*Musca domestica* L., the common house fly, is a vector pest of humans and livestock worldwide; it can carry approximately 100 pathogens and is responsible for several deadly diseases [11,12,13]. *M. domestica* carries pathogens acquired during feeding under unsanitary conditions, transferring these pathogens when it moves to sanitary areas [14,15]. Additionally, *M. domestica* adults cause annoyance to livestock, and *M. domestica* larvae feed voraciously, sometimes resulting in serious injury to affected animals [9]. Cultural practices, chemicals, and biological agents are employed to control and manage *M. domestica* [16]. For example, the adult stage is targeted with synthetic pyrethroid insecticides in dairy facilities and urban environments [17]. Among these pyrethroids, alpha-cypermethrin is commonly used to control dipteran pests, including *M. domestica* [18,19]. However, the over-reliance on alpha-cypermethrin has led to resistance development and increased control costs [20,21]. Indeed, alpha-cypermethrin resistance has been reported in various insect pests worldwide [5,20,21,22,23,24], forcing insecticide users to increase dosages, leading to the aforementioned negative effects.

The risk of insecticide resistance development can be determined through laboratory selection and realized heritability (*h*^2^) estimation, i.e., the fraction of genetic variance to phenotypic variance [25], providing data that help improve resistance management plans and restore insecticide efficacy [10,26]. Resistance to pyrethroids, i.e., lambda-cyhalothrin and permethrin, has been studied in *M. domestica* through laboratory selection and *h*^2^ estimation [6,8]. Additionally, the continuous use of pyrethroids is known to reduce their efficacy in controlling *M. domestica* owing to resistance development and the possibility of cross-resistance (CR) to unexposed insecticides [27]. CR is the phenomenon in which the selection pressure of one insecticide on insect pests favors the development of resistance to other insecticides not used in the field, thereby reducing the effectiveness of several insecticides [2,11,28]. Thus, CR analyses are conducted in pyrethroid-resistant strains of insect pests to inform the rotational use of insecticides [2,11]. Indeed, CR to unexposed pesticides with different or similar modes of action has been studied extensively in various insecticide-resistant *M. domestica* strains [2,8,10,11,29,30,31].

In recent years, low resistance levels to alpha-cypermethrin were observed in Riyadh, Saudi Arabia [5]. However, data on the (1) risk of alpha-cypermethrin resistance, (2) pace at which alpha-cypermethrin resistance changes, and (3) presence or absence of CR are lacking. Such data are crucial for controlling *M. domestica* and insecticide pollution [10,30]. Therefore, the objectives of the present study were to (1) assess the risk of alpha-cypermethrin resistance through laboratory selection of *M. domestica* and *h^2^* estimations, (2) measure the stability of alpha-cypermethrin resistance, and (3) explore the CR phenomenon in alpha-cypermethrin-selected *M. domestica* to inform the rotational use of insecticides.

## 2. Materials and Methods

### 2.1. Chemicals

Fifteen insecticides belonging to pyrethroid, organophosphate, and insect growth regulator classes were used in the bioassays (Table 1).

### 2.2. Collection and Rearing of M. domestica

More than 200 adults of *M. domestica* (both sexes) were trapped in plastic jars (19 × 33 cm) at a dairy farm located in Dirab, Riyadh, Saudi Arabia (24.49° N, 46.60° E). The trapped flies were moved to an aerated cage (40 × 40 cm) in the laboratory and maintained according to the protocol described by Abbas and Hafez [16]. The adult flies were fed from Petri dishes (9 cm in diameter) containing (1) a mixture of powdered milk (1 mg) and sugar (1 mg) and (2) a cotton wick (~3 cm) soaked with deionized water, which were placed in the rearing cages and changed every two days. Plastic cups containing a mixture (500 mL total volume) of wheat bran (20.0 g, Second Milling Company, Riyadh, Saudi Arabia), yeast (5.0 g, S.I. Lesaffre, Marcq-en-Barœul, France), sugar (1.5 g, Al-Osra Company, Jeddah, Saudi Arabia), dry milk powder (1.5 g, Almarai Company, Riyadh, Saudi Arabia), and deionized water (70 mL, Gesellschaft für Labortechnik mbH, Burgwedel, Germany) were also placed in the rearing cages to encourage egg laying and provide larval food. Cups containing eggs were covered with cloth secured by a rubber band to prevent the escape of larvae. Once the larvae had consumed the food provided, fresh food was provided in a glass beaker, and the larvae were allowed to pupate in these beakers. The emerged flies were moved into rearing cages to obtain the next progeny. All *M. domestica* stages were reared under controlled laboratory conditions (27 ± 2 °C, 65 ± 5% relative humidity, and a 12:12 h light:dark photoperiod).

### 2.3. Selection of M. domestica with Alpha-Cypermethrin

The *M. domestica* population collected from the dairy farm, named Field-Pop at generation one (G_1_), was separated into two lines: the alpha-cypermethrin-unselected strain (Alpha-Unsel) was maintained for 24 generations (G_24_) with no chemical treatment in the laboratory; the alpha-cypermethrin-selected strain (Alpha-Sel) was screened continuously with different concentrations of alpha-cypermethrin for 24 generations (Table 2). The first selection was started with the LC_50_ value in the females of the field population and continued for eight generations, until the survival approached approximately 80%. For subsequent generations, the concentrations were increased on the basis of the survival of a sufficient number of adults in the succeeding progeny. On average, 900 adult flies (2–3 days old) in each generation were screened with alpha-cypermethrin through a feeding bioassay [10]. The surviving flies were moved to rearing cages and maintained under the aforementioned laboratory conditions.

### 2.4. Bioassay of Adults

The toxicities of pyrethroid and organophosphate classes against adults were evaluated via a feeding bioassay as described previously by Hafez [5]. Five concentrations (with >0% to <100% mortality) of an insecticide were prepared in 20% sugar solution via serial dilution. Each concentration for each bioassay was replicated three times. For each insecticide, 10, 30, and 150 adults (either males or females) per replicate, concentration, and bioassay were used, respectively. For the control, 10 adults per replicate were used (30 adults in total). The adults were placed in perforated plastic jars (11 cm diameter × 15 cm height), and the mouth of the jar was covered with cloth tightened by rubber bands to prevent the escape of adults. For 2 h before each bioassay, the adults were starved. For each insecticide concentration solution, cotton wicks (~3 cm) were saturated and placed in Petri dishes (9 cm in diameter), which were placed in the plastic jars to feed the starved adults. In the control, cotton wicks saturated with 20% sugar solution only were used. Bioassays were conducted under the aforementioned laboratory conditions. The mortality of adults was assessed after 48 h of exposure owing to the fast action of the tested insecticides [4].

### 2.5. Bioassay of Larvae

The toxicity of insect growth regulators to *M. domestica* larvae was evaluated through a diet incorporation bioassay following Abbas and Hafez [16]. Five concentrations of insect growth regulator (with >0% to <100% mortality) were prepared via serial dilution. For each concentration, 140 mL of insect growth regulator solution was mixed with the larval food consisting of wheat bran (40.0 g), yeast (10.0 g), sugar (3.0 g), and dry milk powder (3.0 g). Each concentration for each bioassay was replicated three times. Second instar larvae were used in the bioassays with 10, 30, and 150 larvae per replicate, concentration, and bioassay, respectively. For the control, the larval diet was mixed with deionized water only, and 3 replicates were used (10 larvae per replicate). Bioassays were performed under the aforementioned laboratory conditions. Mortality was recorded based on the emergence of adults, with unemerged pupae counted as dead [16].

### 2.6. Alpha-Cypermethrin Resistance Stability in M. domestica

The Field-Pop (G_1_) was raised without alpha-cypermethrin selection pressure in the laboratory for 24 generations (G_1_–G_24_) to determine the stability of alpha-cypermethrin resistance. The decline in alpha-cypermethrin resistance (DR) was calculated using the equation of Tabashnik et al. [32]
(1)$$\mathrm{DR}\;=\frac{\left[\log\left({\mathrm{final}\;\mathrm{LC}}_{50}\right)-\log\left({\mathrm{initial}\;\mathrm{LC}}_{50}\right)\right]}{N},$$
where N is the number of generations with no exposure to any chemical. DR ranges from −1 to +1: DR value of −1 illustrates decline in resistance and DR value of +1 illustrates no decline in resistance.

### 2.7. h^2^ Values for Alpha-Cypermethrin Resistance

The *h*^2^ values for alpha-cypermethrin resistance were assessed using the equations of Tabashnik [33] and Abbas et al. [10]
(2)$$h^{2}=\frac{R}{S},$$
where *R* is the alpha-cypermethrin selection response and *S* is the alpha-cypermethrin selection differential. *h*^2^ ranging from 0 to 1: 0 means that most of the differences are not genetic and 1 means that the most of differences are genetic.

*R* was measured using following equation:(3)$$R=\frac{\left[\log\;\left({\mathrm{final}\;\mathrm{LC}}_{50}\;\mathrm{in}\;\mathrm{Alpha}-\mathrm{Sel}\right)-\log\;\left({\mathrm{initial}\;\mathrm{LC}}_{50}\;\mathrm{in}\;\mathrm{Field}-\mathrm{Pop}\right)\right]}{n},$$
where *n* is the total number of generations (G_1_–G_24_) screened with alpha-cypermethrin.

*S* was measured as follows:(4)S=i×σp,
where *i* is the selection intensity (mortality), determined following the method of Tabashnik and McGaughey [34]
(5)$$i=1.583-0.0193336p+0.0000428p^{2}+3.65194/p,$$
where “*p*” is the survival percentage of Alpha-Sel (G_1_–G_24_) screened with alpha-cypermethrin.

*σp* was measured as follows:(6)$$\sigma p=\frac{1}{\mathrm{Average}\;\mathrm{slope}\;\left(G_{1}-G_{24}\right)}.$$

The number of generations (G) required to produce a tenfold increase in the median lethal concentration (LC_50_) of alpha-cypermethrin was determined following Abbas et al. [10].
(7)$$G_{\mathrm{alpha}-\mathrm{cypermethrin}}={\left(h^{2}S\right)}^{-1}$$

Each of *h*^2^, *R*, and *S* were measured in the first phase (G_1_–G_12_) and second phase (G_13_–G_24_) separately (12 generations in each phase) as well as G_1_–G_24_ to determine their changes. Each phase was defined on the basis of half of the total selected generations. The influence of the calculated and assumed slope and *h*^2^ values on alpha-cypermethrin resistance was assessed through G and selection intensity.

### 2.8. Bioassay Data Analyses

To determine the LC_50_, fiducial limits (FLs), chi-square value (χ^2^), and slope (±standard error), the toxicity data of each insecticide were subjected to probit analyses [35] via POLO PLUS Software [36]. The formula of Abbott [37] was considered to correct the mortalities of each bioassay using the mortality of its control treatment. Resistance ratios (RRs) and performance ratios (PRs) were determined using the following equation:(8)$$\frac{{\mathrm{LC}}_{50}\;\mathrm{of}\;\mathrm{an}\;\mathrm{insecticide}\;\mathrm{in}\;\mathrm{Alpha}-\mathrm{Sel}}{{\mathrm{LC}}_{50}\;\mathrm{of}\;\mathrm{an}\;\mathrm{insecticide}\;\mathrm{in}\;\mathrm{Alpha}-\mathrm{Unsel}}$$

The criteria used to classify the RR and PR levels in *M. domestica* were those described by Torres-Vila et al. [38] and Ullah et al. [39], i.e., >100, very high resistance; 31–100, high resistance; 11–30, moderate resistance; 2–10, low resistance; and <2, no resistance.

## 3. Results

### 3.1. Alpha-Cypermethrin Resistance Selection in Alpha-Sel

On average, the survival rate of male and female *M. domestica* was 59.8% and 60.2%, respectively, in G_1_–G_24_ at different alpha-cypermethrin concentrations (Table 2). After continuous laboratory selection, the LC_50_ of alpha-cypermethrin increased from 64.1 ppm (95% FL 51.5–79.8) at G_1_ to 1113.9 ppm (95% FL 741.6–1640.6) at G_24_ in males and from 90.1 ppm (95% FL 46.2–230.2) at G_1_ to 2134.1 ppm (95% FL 1392.7–4225.5) at G_24_ in females. Compared with Alpha-Unsel, the RR for alpha-cypermethrin in G_1_–G_24_ increased from 14.6 to 253.2 in males and from 20.0 to 474.2 in females (Table 3, Figure 1).

### 3.2. Stability of Alpha-Cypermethrin Resistance in M. domestica

After continuous rearing of a Field-Pop without exposure to alpha-cypermethrin, the LC_50_ of alpha-cypermethrin significantly decreased from 64.1 ppm (95% FL 51.5–79.8) at G_1_ to 4.4 ppm (95% FL 3.2–5.6) at G_24_ in males and from 90.1 ppm (95% FL 46.2–230.2) at G_1_ to 4.5 ppm (95% FL 3.1–5.9) at G_24_ in females. The DR to alpha-cypermethrin was from –0.1 (G_5_) to –0.05 (G_24_) in both *M. domestica* sexes without insecticide exposure for 24 generations (Table 4).

### 3.3. h^2^ of Alpha-Cypermethrin Resistance in M. domestica

In *M. domestica* females and males, the overall *h*^2^ values of alpha-cypermethrin resistance in G_1_–G_24_ were 0.18 and 0.17, respectively. In females, in the first (G_1_–G_12_) and second (G_13_–G_24_) phases of selection, the estimated *h*^2^ was 0.15 and 0.24, respectively (Table 5). In males, in G_1_–G_12_ and G_13_–G_24_, the estimated *h*^2^ was 0.14 and 0.20, respectively (Table 5).

### 3.4. Projected Rate of Alpha-Cypermethrin Resistance Development

With each selection causing 10–90% mortality for female *M. domestica*, the generations required for a tenfold increase in the LC_50_ of alpha-cypermethrin were 6.3–48.3, 4.1–31.0, and 3.0–22.9, given *h*^2^ values of 0.18, 0.28, and 0.38, respectively, and a constant slope of 2.0 (Figure 2A). For male *M. domestica* at similar selection intensities, the G required for a tenfold increase in the LC_50_ of alpha-cypermethrin was 7.0–53.7, 4.4–33.8, and 3.2–24.7, given *h*^2^ values of 0.17, 0.27, and 0.37, respectively, and a constant slope of 2.1 (Figure 2B).

At a *h*^2^ value of 0.18 and with slopes of 2.0, 3.0, and 4.0, G values of 6.3–48.3, 9.5–72.4, and 12.6–96.5, respectively, were required for a tenfold increase in the LC_50_ of alpha-cypermethrin in females (Figure 3A). For males, G values of 7.0–53.7, 10.4–79.2, and 13.7–104.8 were required for a tenfold increase in the LC_50_ of alpha-cypermethrin, given a constant *h*^2^ of 0.17 and slopes of 2.1, 3.1, and 4.1, respectively (Figure 3B). These results indicate that fluctuations in *h*^2^ and slope cause variation in the alpha-cypermethrin resistance development rate.

### 3.5. Cross-Resistance (CR) Patterns

Compared with Alpha-Unsel, Alpha-Sel (G_24_) exhibited (1) moderate CR between alpha-cypermethrin and bifenthrin (PR = 15.5-fold), deltamethrin (PR = 28.4-fold), or cyfluthrin (16.8-fold); (2) low CR between alpha-cypermethrin and cypermethrin (PR = 5.0-fold), fenitrothion (PR = 6.1-fold), chlorpyrifos (PR = 4.8-fold), malathion (PR = 2.1-fold), diazinon (PR = 8.1-fold), pirimiphos-methyl (PR = 2.2-fold), triflumuron (PR = 3.3-fold), or pyriproxyfen (PR = 5.0-fold); and (3) no CR between alpha-cypermethrin and diflubenzuron (PR = 1.3-fold), cyromazine (PR = 1.9-fold), or methoxyfenozide (PR = 1.2-fold) (Table 6).

## 4. Discussion

Hafez previously found low resistance (2- to 4-fold) to alpha-cypermethrin in *M. domestica* females and almost no resistance to low resistance (0.5- to 7.0-fold) in *M. domestica* males [5]. However, in the present study, the reselection of *M. domestica* adults with alpha-cypermethrin for 24 generations increased resistance by 253.2- and 474.2-fold in males and females, respectively. Therefore, *M. domestica* adults can quickly gain very high resistance to alpha-cypermethrin after continuous exposure in the laboratory. Similarly, high resistance to other pyrethroids, including lambda-cyhalothrin, deltamethrin, and permethrin, has been found in *M. domestica* [6,8,40]. Nevertheless, *M. domestica* populations collected from Saudi dairies exhibited little or no field-evolved resistance [5], although inappropriate pesticide use in these dairies could lead to the development of alpha-cypermethrin resistance in *M. domestica*. Indeed, the present selection experiment revealed that alpha-cypermethrin selection pressure markedly affected the field population, leading to the rapid development of resistance after 24 generations. In addition to *M. domestica*, alpha-cypermethrin resistance had been found worldwide in pests such as *Anopheles stephensi* Liston [41], *Rhipicephalus microplus* Canestrini [42], *Bactrocera oleae* Rossi. [20,22], *Blattella germanica* L. [43], and *Stomoxys calcitrans* L. [24].

Estimating *h*^2^ using a quantitative genetic model can support predictions of variation in a specific trait (e.g., insecticide resistance) when the variation is genetically linked to the trait. The expression of such traits depends on the nature of resistance genes and environmental factors [10,33], and the rate of developing resistance is directly proportional to the *h*^2^ value of any insecticide [34]. A high *h*^2^ value indicates a higher risk of genetic resistance development because more resistance genes are inherited by the next generation [44]. In contrast, a low *h*^2^ value indicates higher phenotypic variation that may arise from gene mutation, population migration, selection pressure, insecticide rotation, and environmental influences under laboratory and field conditions [45]. In the present study, the low *h*^2^ values of 0.18 and 0.17 for female and male *M. domestica*, respectively, indicate low genetic variation and high phenotypic variation, i.e., *M. domestica* exhibited a low probability of developing genetic resistance to alpha-cypermethrin. Previous studies have also found low *h*^2^ values in insecticide-selected *M. domestica* strains, e.g., 0.07 for lambda-cyhalothrin [6], 0.23 for permethrin [8], 0.05 for fipronil [9], 0.17 for methoxyfenozide [46], 0.03 for pyriproxyfen [47], 0.02 for flonicamid [10], and 0.08 for diflubenzuron [30]. In the current study, *R* and *S* declined as the alpha-cypermethrin selection pressure was increased, producing a lower *h*^2^ in the first half of alpha-cypermethrin selection than in the second half. Therefore, the alleles responsible for developing alpha-cypermethrin resistance existed at low levels in the first half of selection, whereas these levels increased after further alpha-cypermethrin exposure in the second half of selection. Random drift might also explain these observations. The present results contrast with those of Abbas and Shad [6] and Khan [8], who found additive genetic changes in the first half of selection that decreased in the second half of selection after further exposure of *M. domestica* strains, although our results are similar to those of Shah et al. [47] and Abbas et al. [9]. The *h*^2^ of insecticide resistance might fluctuate because of changes in allele frequency and the environment over time [33]; consequently, forecasts based on *h*^2^ estimation in laboratory-selected strains must be interpreted prudently in relation to *M. domestica* management. However, the conditions in the field are not counterparts to the controlled conditions in a laboratory, although the estimated *h*^2^ of alpha-cypermethrin resistance mediated with laboratory selection has implications for resistance management programs [10,34]. The lower *h*^2^ value in this study reveals that many generations may be needed before *M. domestica* reaches a significant resistance level, although, alpha-cypermethrin should be used rotationally for controlling this pest specie to prolong its usefulness.

Estimating the rate of resistance development through the number of generations (G = *h*^2^S^−1^) is a valuable step toward establishing rational resistance management strategies for insect vectors [6]. Such estimated rates have been determined previously in *M. domestica* strains selected with lambda-cyhalothrin [6], permethrin [8], flonicamid [10], clothianidin [31], and diflubenzuron [30]. According to the results in Figure 1 and Figure 2, the risk of *M. domestica* males and females developing alpha-cypermethrin resistance increases when the *h*^2^ value is increased. This reveals that as the *h*^2^ value increases, the number of generations needed for a ten-fold increase in alpha-cypermethrin resistance decreases. Therefore, the populations with a high *h*^2^ may become resistant after few generations when exposed to intense selection pressure of insecticide under field conditions. Therefore, alpha-cypermethrin should be applied prudently for the control of *M. domestica*.

The instability of resistance to any insecticide is essential for its prolonged potency, and determining this instability is useful for developing effective resistance management strategies. For instance, when insecticide resistance is unstable, the potency of a specific insecticide may persist if it is rotated with another insecticide. However, when insecticide resistance is stable, the insecticide should not be included in insecticide resistance management plans to avoid resistance complications [48,49,50]. The present results indicated that alpha-cypermethrin resistance was unstable in an *M. domestica* population collected from a Saudi dairy farm and reared for 24 generations without insecticide exposure. Indeed, the LC_50_ values decreased greatly from G_1_ to G_24_ (from 64.1 to 4.4 ppm in males and from 90.1 to 4.5 ppm in females). This indicates that the mechanism of alpha-cypermethrin resistance is unstable and the unstable resistant alleles require higher fitness costs for their development and survival [51], so the population reverts towards susceptibility after 24 generations. Genetic drift and gene mutation due to stop of selection pressure might be the other reasons for instability of alpha-cypermethrin resistance [32]. Similarly, unstable insecticide resistance to lambda-cyhalothrin was found previously in *M. domestica* [52]. In contrast, stable resistance to permethrin was found previously in *M. domestica* [8].

Information on CR is required to choose alternative insecticides for rational management programs [10,30]. The present CR bioassay results revealed moderate CR between alpha-cypermethrin and bifenthrin, deltamethrin, or cyfluthrin, low CR between alpha-cypermethrin and cypermethrin, fenitrothion, chlorpyrifos, malathion, diazinon, pirimiphos-methyl, triflumuron, or pyriproxyfen, and no CR between alpha-cypermethrin and diflubenzuron, cyromazine, or methoxyfenozide in Alpha-Sel *M. domestica*. CR between alpha-cypermethrin and bifenthrin, deltamethrin, cyfluthrin, or cypermethrin was expected in Alpha-Sel *M. domestica* as these insecticides have a similar mode of action, a sodium channel modulator. However, the low CR of cypermethrin is interesting in the Alpha-Sel, despite the same molecular target and mode of action. CR between alpha-cypermethrin and the organophosphates (fenitrothion, chlorpyrifos, malathion, diazinon, and pirimiphos-methyl) and insect growth regulators (triflumuron, diflubenzuron, pyriproxyfen, cyromazine, and methoxyfenozide) was not expected as they are in different chemical classes and possess different modes of action [53]. The present results suggest that the no or low CR of organophosphates and insect growth regulators with alpha-cypermethrin may be due to differences in their mode of actions and lower metabolic detoxification. In a previous study, lambda-cyhalothrin-selected *M. domestica* exhibited very low CR with indoxacarb and abamectin and no CR with bifenthrin and methomyl [2]. In addition, permethrin-selected *M. domestica* exhibited low CR with 𝛽-cyfluthrin and deltamethrin and no CR with imidacloprid and spinosad [8]. Similarly, a thiamethoxam-resistant strain of *Aphis gossypii* Glover exhibited low CR with alpha-cypermethrin [54]. In contrast, permethrin-resistant *Aedes aegypti* L. exhibited high CR with deltamethrin, lambda-cyhalothrin, cypermethrin, alpha-cypermethrin, and zeta-cypermethrin but no CR with bifenthrin [55]. Conversely, diflubenzuron-resistant *M. domestica* exhibited no CR with alpha-cypermethrin, cypermethrin, bifenthrin, deltamethrin, cyfluthrin, malathion, pyriproxyfen, and methoxyfenozide [30]. Given the lack of or low CR between alpha-cypermethrin and cypermethrin, fenitrothion, chlorpyrifos, malathion, diazinon, pirimiphos-methyl, triflumuron, pyriproxyfen, diflubenzuron, cyromazine, or methoxyfenozide, these insecticides likely represent good alternatives to alpha-cypermethrin for controlling *M. domestica*.

## 5. Conclusions

In summary, Alpha-Sel *M. domestica* exhibited rapid development of alpha-cypermethrin resistance under continuous selection pressure in the laboratory, which may reflect the possibility of alpha-cypermethrin resistance development in this pest species if the insecticide is applied continuously for a long period in Saudi dairies or urban settings. However, the low *h*^2^ values observed in the present study are encouraging in terms of establishing resistance management programs for alpha-cypermethrin and prolonging its potency against *M. domestica*. In addition, the absence of or low CR between alpha-cypermethrin and eleven other insecticides (cypermethrin, fenitrothion, chlorpyrifos, malathion, diazinon, pirimiphos-methyl, triflumuron, pyriproxyfen, diflubenzuron, cyromazine, or methoxyfenozide) indicates that several options are available for insecticide rotation in *M. domestica* control strategies.

## Figures and Tables

**Figure 1 insects-14-00233-f001:**
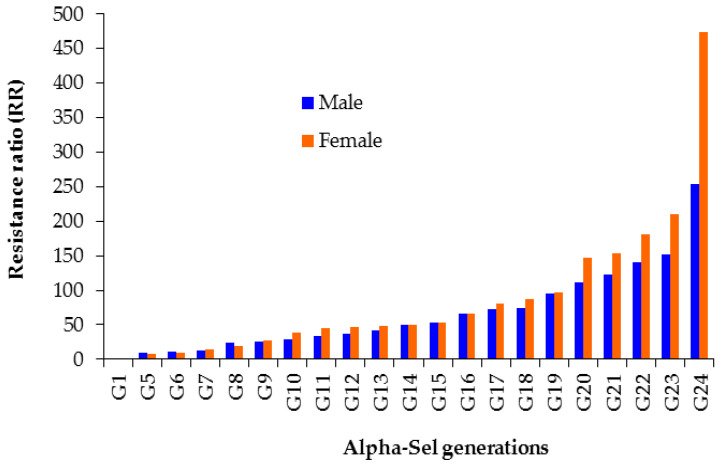
The trend in alpha-cypermethrin resistance development in Alpha-Sel each generation compared to respective Alpha-Unsel generation. Resistance ratio (RR) = LC_50_ for alpha-cypermethrin in Alpha-Sel each generation (G_1_–G_24_)/LC_50_ of alpha-cypermethrin in Alpha-Unsel respective generation (G_1_–G_24_).

**Figure 2 insects-14-00233-f002:**
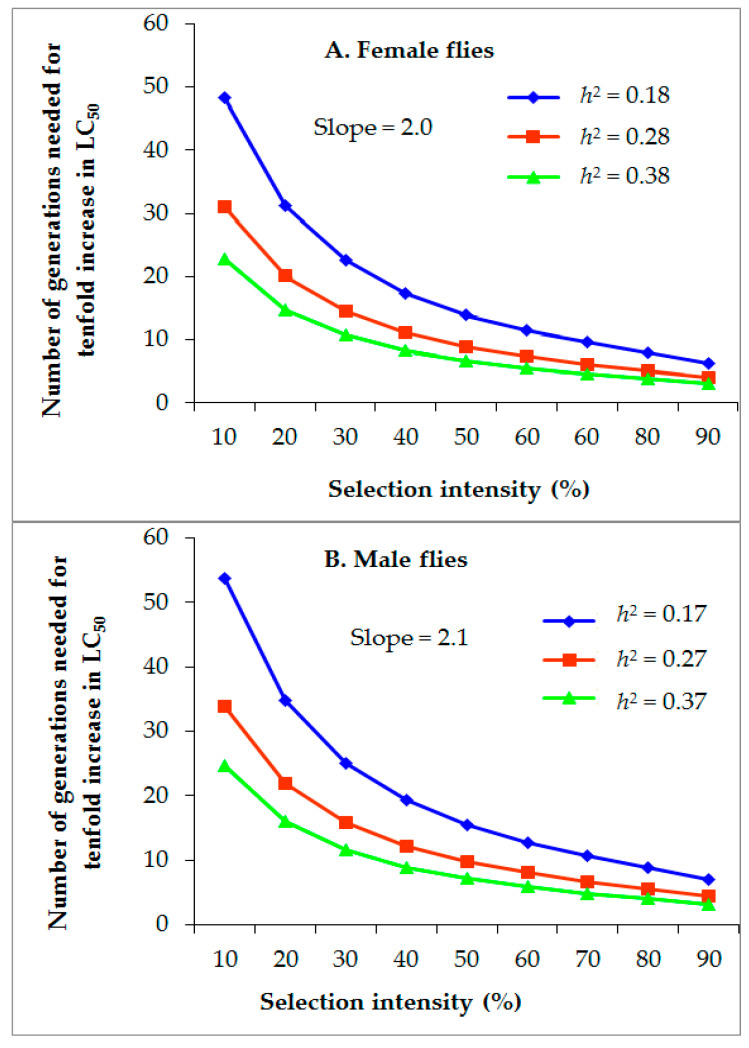
Effect of different heritability values on the number of *M. domestica* generations required to increase the LC_50_ of alpha-cypermethrin by tenfold at different selection intensities, (**A**) = female flies and (**B**) = male flies. For each symbol, *h*^2^ represents realized heritability at the given value.

**Figure 3 insects-14-00233-f003:**
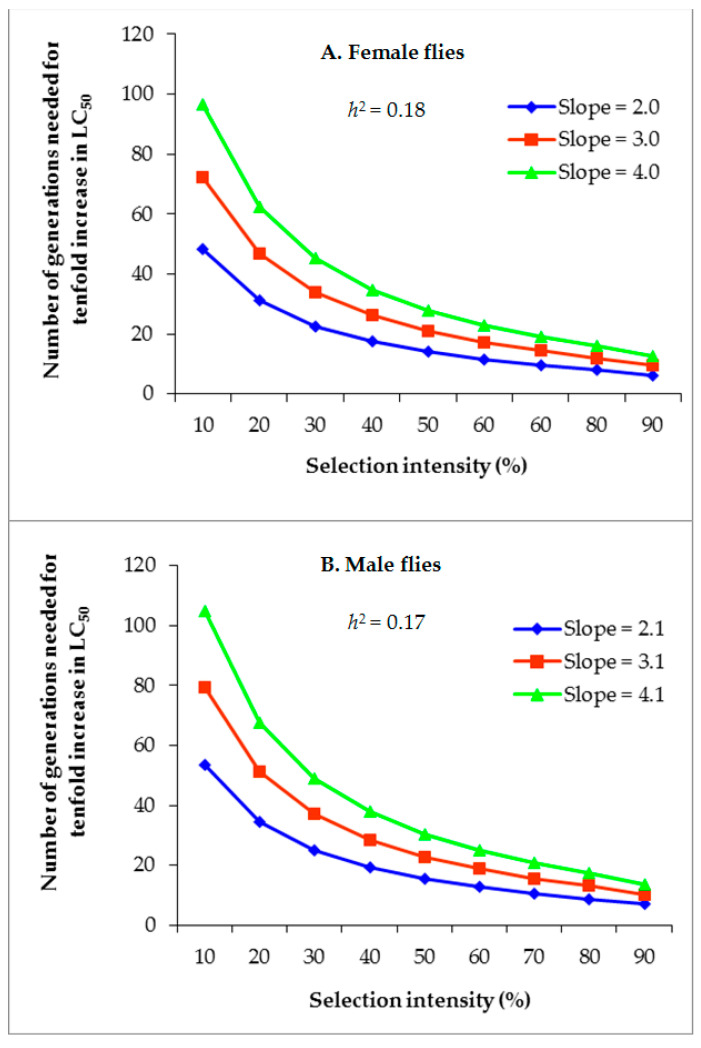
Effect of different slope values on the number of *M. domestica* generations required to increase the LC_50_ of alpha-cypermethrin by tenfold at different selection intensities, (**A**) = female flies and (**B**) = male flies.

**Table 1 insects-14-00233-t001:** Insecticides used in the *Musca domestica* bioassays.

IRAC ^†^ Chemical Class	Active Ingredient	Primary Site of Action	Trade Name	Formulation (%)	Manufacturer
2A Organophosphate	Fenitrothion	Acetyl cholinesterase inhibitors	Fentox	50% EC	Pioneers Chemicals Factory Co., Saudi Arabia
	Chlorpyrifos		Chlorfet	48% EC	Masani Chemicals, Jordan
	Malathion		Delthion	57% EC	Saudi Delta Company, Saudi Arabia
	Diazinon		Diazinon	60% EC	APCO, Saudi Arabia
	Pirimiphos-methyl		Actikil	50% EC	Astrachem, Saudi Arabia
3A Pyrethroid	Alpha-cypermethrin	Sodium channel modulators	Alphaquest	10% EC	Astrachem, Saudi Arabia
	Cypermethrin		Montothrin	10% EC	Montajat Agrochemicals, Saudi Arabia
	Bifenthrin		Biflex	8% SC	FMC, Belgium
	Deltamethrin		K-Othrine	25% SC	Bayer Crop Sciences, France
	Cyfluthrin		Solfac	5% EW	Bayer Crop Sciences, Germany
7C Pyriproxyfen	Pyriproxyfen	Juvenile hormone mimics	Admiral	10% EC	Sumitomo Chemicals, Japan
15 Benzoylureas	Diflubenzuron	Inhibitors of chitin biosynthesis affecting CHS1	Diflon	25% WP	Saudi Delta Company, Saudi Arabia
	Triflumuron		Starycide	48% SC	Bayer Crop Sciences, Germany
17 Cyromazine	Cyromazine	Molting disruptors, Diptera	Novasat	75% WP	Astranova Chemicals, Saudi Arabia
18 Diacylhydrazines	Methoxyfenozide	Ecdysone receptor agonists	Runner	24% SC	Dow Agro Sciences, United Kingdom

^†^ Insecticide Resistance Action Committee.

**Table 2 insects-14-00233-t002:** Screening history of alpha-cypermethrin-selected strain of *Musca domestica*.

Concentration (ppm)	Generation	Number of Males Exposed	Survival (%)	Number of Females Exposed	Survival (%)
90	G_1_	435	16.6	352	36.4
90	G_2_	287	23.3	278	33.5
90	G_3_	642	28.0	536	40.1
90	G_4_	994	35.9	338	37.9
90	G_5_	993	46.2	712	45.8
90	G_6_	193	52.8	207	62.8
90	G_7_	142	64.1	48	64.6
90	G_8_	279	84.6	146	76.7
372	G_9_	642	28.0	536	40.1
372	G_10_	75	50.7	41	36.6
372	G_11_	268	48.1	151	40.4
372	G_12_	366	80.9	163	36.2
372	G_13_	141	64.5	79	57.0
372	G_14_	227	62.6	208	60.6
372	G_15_	622	66.2	637	75.4
372	G_16_	705	85.8	656	84.8
372	G_17_	815	89.0	680	86.8
1000	G_18_	573	62.5	663	71.2
1000	G_19_	428	77.1	443	80.4
1000	G_20_	546	71.4	465	80.6
1000	G_21_	659	77.2	694	81.0
1000	G_22_	656	80.9	707	80.6
1000	G_23_	613	78.3	695	74.8
Average			59.8		60.2

Survival data were not recorded at G_24_ but were selected with 1000 ppm.

**Table 3 insects-14-00233-t003:** Toxicity and development of resistance to alpha-cypermethrin in *Musca domestica*.

Strain (Generation)	Males					Females				
	LC_50_ (95% FL) ^†^ (ppm)	Slope ± SE	*χ*^2^ (df)	*P*	RR ^‡^	LC_50_ (95% FL) ^†^ (ppm)	Slope ± SE	*χ*^2^ (df)	*p*	RR ^‡^
Alpha-Unsel (G_24_)	4.4 (3.2–5.6)	2.6 ± 0.4	0.6 (3)	0.9	1.0	4.5 (3.1–5.9)	2.0 ± 0.3	2.3 (3)	0.5	1.0
Field-Pop (G_1_)	64.1 (51.5–79.8)	2.7 ± 0.4	1.8 (3)	0.6	14.6	90.1 (46.2–230.2) *	2.4 ± 0.4	6.8 (3)	0.1	20.0
Alpha-Sel (G_5_)	180.4 (136.3–232.2)	2.5 ± 0.4	2.3 (3)	0.5	41.0	208.8 (146.6–278.6)	2.4 ± 0.4	1.2 (3)	0.8	46.4
Alpha-Sel (G_6_)	201.2 (90.0–548.2)	1.5 ± 0.3	4.2 (3)	0.2	45.7	239.9 (175.5–346.5)	1.6 ± 0.3	1.8 (3)	0.6	53.3
Alpha-Sel (G_7_)	204.3 (155.0–267.2)	2.3 ± 0.4	2.7 (3)	0.4	46.4	283.5 (186.8–407.4)	2.0 ± 0.4	2.8 (3)	0.4	63.0
Alpha-Sel (G_8_)	316.3 (128.7–723.2)	1.9 ± 0.3	5.8 (3)	0.1	71.9	332.0 (254.3–420.8)	2.3 ± 0.3	2.2 (3)	0.5	73.8
Alpha-Sel (G_9_)	316.5 (148.3–478.2)	1.4 ± 0.3	0.6 (3)	0.9	71.9	393.0 (244.4–609.2)	2.4 ± 0.3	3.5 (3)	0.3	87.3
Alpha-Sel (G_10_)	319.3 (156.9–610.7)	1.9 ± 0.3	4.3 (3)	0.2	72.6	467.7 (259.2–1021.3)	1.9 ± 0.3	4.4 (3)	0.2	103.9
Alpha-Sel (G_11_)	352.4 (182.8–515.4)	1.5 ± 0.3	0.8 (3)	0.8	80.1	495.9 (199.5–1067.3)	2.0 ± 0.4	3.9 (3)	0.3	110.2
Alpha-Sel (G_12_)	367.3 (161.4–889.7)	1.5 ± 0.3	3.9 (3)	0.3	83.5	497.0 (315.1–686.0)	1.6 ± 0.3	1.2 (3)	0.8	110.4
Alpha-Sel (G_13_)	383.4 (277.5–504.9)	2.4 ± 0.4	2.7 (3)	0.4	87.1	500.0 (376.7–695.0)	1.8 ± 0.3	0.6 (3)	0.9	111.1
Alpha-Sel (G_14_)	410.3 (333.2–503.1)	3.0 ± 0.4	1.5 (3)	0.7	93.3	504.3 (375.7–714.1)	1.8 ± 0.3	1.1 (3)	0.8	112.1
Alpha-Sel (G_15_)	429.5 (117.7–666.3)	2.9 ± 0.6	3.1 (3)	0.4	97.6	527.7 (409.6–682.2)	2.2 ± 0.3	0.4 (3)	0.9	117.3
Alpha-Sel (G_16_)	440.8 (245.0–636.9)	1.4 ± 0.3	1.1 (3)	0.8	100.2	596.5 (389.5–823.8)	1.6 ± 0.3	0.1 (3)	0.9	132.6
Alpha-Sel (G_17_)	480.7 (345.4–628.6)	2.6 ± 0.4	1.1 (3)	0.8	109.3	649.1 (404.2–930.1)	1.4 ± 0.3	0.3 (3)	0.9	144.2
Alpha-Sel (G_18_)	486.6 (294.8–683.9)	1.5 ± 0.3	0.3 (3)	1.0	110.6	684.6 (545.7–877.0)	2.5 ± 0.4	0.6 (3)	0.9	152.1
Alpha-Sel (G_19_)	564.3 (382.1–855.0)	3.8 ± 0.5	4.5 (3)	0.2	128.3	740.5 (565.5–1019.1)	2.0 ± 0.3	2.2 (3)	0.5	164.6
Alpha-Sel (G_20_)	579.9 (357.6–821.2)	1.6 ± 0.3	0.6 (3)	0.9	131.8	967.5 (674.3–1379.5)	1.5 ± 0.3	0.1 (3)	0.9	215.0
Alpha-Sel (G_21_)	614.6 (218.2–1225.2)	2.2 ± 0.4	4.1 (3)	0.3	139.7	983.3 (692.5–1350.7)	1.8 ± 0.3	0.6 (3)	0.9	218.5
Alpha-Sel (G_22_)	688.4 (460.2–955.4)	1.5 ± 0.3	0.7 (3)	0.9	156.5	1122.6 (726.7–1642.6)	1.7 ± 0.3	0.3 (3)	0.9	249.5
Alpha-Sel (G_23_)	743.3 (343.1–1329.9)	1.7 ± 0.3	3.6 (3)	0.3	168.9	1137.5 (848.4–1560.4)	1.8 ± 0.3	2.4 (3)	0.5	252.8
Alpha-Sel (G_24_)	1113.9 (741.6–1640.6)	1.6 ± 0.3	0.7 (3)	0.9	253.2	2134.1 (1392.7–4225.5)	2.4 ± 0.4	3.4 (3)	0.3	474.2

Bioassays were not performed for G_2_–G_4_. * Published results (Hafez 2021). ^†^ Median lethal concentration. FL = Fiducial limit. SE = standard error. ^‡^ Resistance ratio (was calculated as LC_50_ for alpha-cypermethrin in Alpha-Sel/LC_50_ of alpha-cypermethrin in Alpha-Unsel (G_24_).

**Table 4 insects-14-00233-t004:** Stability of alpha-cypermethrin resistance in a field strain of *Musca domestica*.

Strain (Generation)	Males						Females					
	LC_50_ (95% FL) ^†^ (ppm)	Slope ± SE	*χ*^2^ (df)	*P*	RR ^‡^	DR	LC_50_ (95% FL) ^†^ (ppm)	Slope ± SE	*χ*^2^ (df)	*p*	RR ^‡^	DR
Field-Pop (G_1_)	64.1 (51.5–79.8)	2.7 ± 0.4	1.8 (3)	0.6	14.6		90.1 (46.2–230.2)	2.4 ± 0.4	6.8 (3)	0.1	20.0	
Alpha-Unsel (G_5_)	20.5 (15.4–27.2)	1.9 ± 0.3	2.0 (3)	0.6	4.7	−0.10	27.3 (20.7–36.9)	1.9 ± 0.3	0.7 (3)	0.9	6.1	−0.10
Alpha-Unsel (G_6_)	17.5 (12.2–24.1)	1.6 ± 0.3	2.4 (3)	0.5	4.0	−0.09	24.9 (17.1–35.5)	1.7 ± 0.3	1.9 (3)	0.6	5.5	−0.09
Alpha-Unsel (G_7_)	16.6 (8.0–30.0)	2.1 ± 0.3	4.8 (3)	0.2	3.8	−0.08	20.6 (15.3–27.6)	1.8 ± 0.3	0.7 (3)	0.9	4.6	−0.09
Alpha-Unsel (G_8_)	13.1 (9.2–17.2)	2.7 ± 0.5	2.0 (3)	0.6	3.0	−0.09	17.3 (12.5–23.2)	1.8 ± 0.3	1.1 (3)	0.8	3.8	−0.09
Alpha-Unsel (G_9_)	12.2 (8.4–16.2)	1.9 ± 0.3	1.6 (3)	0.7	2.8	−0.08	14.0 (9.6–18.1)	2.7 ± 0.6	2.9 (3)	0.4	3.1	−0.09
Alpha-Unsel (G_10_)	10.8 (6.6–17.8)	2.6 ± 0.4	4.3 (3)	0.2	2.5	−0.08	12.1 (7.9–19.2)	2.8 ± 0.4	4.0 (3)	0.3	2.7	−0.09
Alpha-Unsel (G_11_)	10.5 (4.9–18.2)	2.0 ± 0.4	3.3 (3)	0.4	2.4	−0.07	11.2 (8.6–14.6)	2.1 ± 0.3	1.4 (3)	0.7	2.5	−0.08
Alpha-Unsel (G_12_)	9.8 (7.8–12.4)	2.4 ± 0.3	1.2 (3)	0.8	2.2	−0.07	10.6 (8.2–13.7)	2.2 ± 0.3	2.7 (3)	0.4	2.4	−0.08
Alpha-Unsel (G_13_)	9.3 (7.0–11.8)	2.8 ± 0.5	1.5 (3)	0.7	2.1	−0.06	10.5 (8.0–13.3)	2.7 ± 0.4	0.7 (3)	0.9	2.3	−0.07
Alpha-Unsel (G_14_)	8.3 (4.0–14.4)	1.9 ± 0.3	3.8 (3)	0.3	1.9	−0.06	10.2 (7.7–13.5)	2.0 ± 0.3	2.9 (3)	0.4	2.3	−0.07
Alpha-Unsel (G_15_)	8.2 (3.4–15.9)	1.9 ± 0.3	5.0 (3)	0.2	1.9	−0.06	9.9 (6.7–13.7)	1.8 ± 0.3	0.2 (3)	1.0	2.2	−0.06
Alpha-Unsel (G_16_)	6.7 (2.1–13.8)	1.4 ± 0.3	3.8 (3)	0.3	1.5	−0.06	9.1 (6.7–12.4)	1.7 ± 0.3	0.5 (3)	0.9	2.0	−0.06
Alpha-Unsel (G_17_)	6.6 (4.9–8.4)	2.3 ± 0.3	1.9 (3)	0.6	1.5	−0.06	8.0 (5.5–10.6)	2.4 ± 0.4	2.4 (3)	0.5	1.8	−0.06
Alpha-Unsel (G_18_)	6.6 (4.5–9.0)	1.6 ± 0.3	0.3 (3)	1.0	1.5	−0.05	7.8 (5.5–10.8)	1.6 ± 0.3	0.2 (3)	1.0	1.7	−0.06
Alpha-Unsel (G_19_)	5.9 (4.0–8.0)	1.7 ± 0.3	0.6 (3)	0.9	1.3	−0.05	7.6 (5.2–10.5)	1.5 ± 0.3	0.3 (3)	1.0	1.7	−0.06
Alpha-Unsel (G_20_)	5.2 (3.7–6.8)	2.0 ± 0.3	1.2 (3)	0.8	1.2	−0.05	6.6 (4.7–8.7)	1.9 ± 0.3	1.3 (3)	0.7	1.5	−0.06
Alpha-Unsel (G_21_)	5.0 (3.4–6.5)	2.6 ± 0.4	0.8 (3)	0.9	1.1	−0.05	6.4 (4.2–8.9)	1.9 ± 0.3	0.2 (3)	1.0	1.4	−0.05
Alpha-Unsel (G_22_)	4.9 (3.4–6.6)	1.9 ± 0.3	0.4 (3)	0.9	1.1	−0.05	6.2 (4.5–8.2)	1.9 ± 0.3	0.2 (3)	1.0	1.4	−0.05
Alpha-Unsel (G_23_)	4.9 (3.2–6.7)	2.2 ± 0.4	2.2 (3)	0.5	1.1	−0.05	5.4 (3.8–7.1)	1.9 ± 0.3	1.1 (3)	0.8	1.2	−0.05
Alpha-Unsel (G_24_)	4.4 (3.2–5.6)	2.6 ± 0.4	0.6 (3)	0.9	1.0	−0.05	4.5 (3.1–5.9)	2.0 ± 0.3	2.3 (3)	0.5	1.0	−0.05

Bioassays were not performed for G_2_–G_4_. ^†^ Median lethal concentration. FL = Fiducial limit. SE = Standard error. ^‡^ Resistance ratio [was calculated as LC_50_ for alpha-cypermethrin in Alpha-Unsel (G_1_–G_24_)/LC_50_ of alpha-cypermethrin in Alpha-Unsel (G_24_)]. DR = Decline in alpha-cypermethrin resistance.

**Table 5 insects-14-00233-t005:** Realized heritability (*h*^2^) for alpha-cypermethrin resistance in Alpha-Sel *Musca domestica*.

Generation	Insecticide	Initial LC_50_ (log) (ppm)	Final LC_50_ (log) (ppm)	G ^1^	*R* ^2^	*p* ^3^	*i* ^4^	Mean Slope	*σ* *p* ^5^	*S* ^6^	*h* ^2^
Female flies											
12 (G_1_–G_12_)	Alpha-cypermethrin	90.1 (2.0)	497.0 (2.7)	12	0.06	45.9	0.87	2.1	0.48	0.41	0.15
12 (G_13_–G_24_)	Alpha-cypermethrin	500.0 (2.7)	2134.1 (3.3)	12	0.05	75.7	0.41	1.9	0.53	0.22	0.24
24 (G_1_–G_24_)	Alpha-cypermethrin	90.1 (2.0)	2134.1 (3.3)	24	0.06	60.2	0.63	2.0	0.50	0.32	0.18
Male flies											
12 (G_1_–G_12_)	Alpha-cypermethrin	64.1 (1.81)	367.3 (2.57)	12	0.06	46.6	0.85	1.9	0.53	0.45	0.14
12 (G_13_–G_24_)	Alpha-cypermethrin	383.4 (2.58)	1113.9 (3.05)	12	0.04	74.1	0.43	2.2	0.45	0.20	0.20
24 (G_1_–G_24_)	Alpha-cypermethrin	64.1 (1.81)	1113.9 (3.05)	24	0.05	59.8	0.64	2.1	0.48	0.31	0.17

^1^ Number of generations screened with alpha-cypermethrin. ^2^ Selection response. ^3^ Mean surviving males and females in selection. ^4^ Intensity of selection. ^5^ Phenotypic variation. ^6^ Selection differential. *h*^2^ = Realized heritability of alpha-cypermethrin resistance.

**Table 6 insects-14-00233-t006:** Cross-resistance to different insecticides in alpha-cypermethrin–resistant *Musca domestica*.

Strain	Insecticide	Conc. (ppm)	LC_50_ (95% FL) ^†^ (ppm)	Slope ± SE	χ^2^ (df)	*p*	PR ^‡^
Alpha-Unsel (G_24_)	Bifenthrin	128–2048	254.8 (84.9–435.7)	0.9 ± 0.3	0.03 (3)	1.0	1.0
	Deltamethrin	128–2048	146.5 (45.0–247.3)	1.1 ± 0.3	1.0 (3)	0.8	1.0
	Cyfluthrin	128–2048	139.3 (36.9–243.4)	1.1 ± 0.3	0.5 (3)	0.9	1.0
	Cypermethrin	128–2048	172.6 (60.0–283.8)	1.1 ± 0.3	0.3 (3)	0.9	1.0
	Fenitrothion	128–2048	140.1 (54.0–226.9)	0.9 ± 0.2	0.3 (3)	1.0	1.0
	Malathion	128–2048	213.8 (62.5–368.8)	0.9 ± 0.3	0.2 (3)	1.0	1.0
	Diazinon	2–32	3.0 (1.9–4.1)	1.9 ± 0.3	1.9 (3)	0.6	1.0
	Pirimiphos-methyl	128–2048	153.8 (68.4–236.0)	1.4 ± 0.3	0.3 (3)	1.0	1.0
	Chlorpyrifos	32–512	42.3 (18.6–65.3)	1.3 ± 0.3	4.9 (3)	0.2	1.0
	Diflubenzuron	0.25–4	0.7 (0.4–1.1)	2.2 ± 0.3	3.6 (3)	0.3	1.0
	Triflumuron	0.125–2	0.2 (0.1–0.3)	1.7 ± 0.3	4.4 (3)	0.2	1.0
	Pyriproxyfen	0.0078–0.125	0.01 (0.01–0.02)	2.0 ± 0.4	2.0 (3)	0.6	1.0
	Cyromazine	0.125–2	0.4 (0.3–0.5)	2.3 ± 0.3	0.5 (3)	0.9	1.0
	Methoxyfenozide	4–64	10.1 (7.6–12.9)	2.2 ± 0.3	0.7 (3)	0.9	1.0
Alpha-Sel (G_24_)	Bifenthrin	625–10000	3941.5 (2012.4–12551.0)	1.7 ± 0.3	4.6 (3)	0.2	15.5
	Deltamethrin	512–8192	4158.2 (3186.6–5879.9)	2.1 ± 0.3	2.5 (3)	0.5	28.4
	Cyfluthrin	562.5–9000	2336.6 (1559.3–3532.1)	1.3 ± 0.3	0.9 (3)	0.8	16.8
	Cypermethrin	562.5–9000	867.0 (671.4–1061.5)	3.3 ± 0.6	2.7 (3)	0.4	5.0
	Fenitrothion	128–2048	860.5 (622.8–1317.7)	1.6 ± 0.3	0.5 (3)	0.9	6.1
	Chlorpyrifos	128–2048	201.6 (42.8–362.3)	1.5 ± 0.3	3.3 (3)	0.4	4.8
	Malathion	128–2048	437.5 (293.7–625.7)	1.4 ± 0.3	1.9 (3)	0.6	2.1
	Diazinon	16–256	24.1 (5.2–43.5)	0.9 ± 0.3	0.1 (3)	1.0	8.1
	Pirimiphos-methyl	128–2048	332.1 (207.7–474.8)	1.4 ± 0.3	1.1 (3)	0.8	2.2
	Triflumuron	0.25–4	0.7 (0.4–1.0)	1.2 ± 0.3	0.4 (3)	0.9	3.3
	Diflubenzuron	0.25–4	0.9 (0.5–1.4)	2.8 ± 0.4	4.2 (3)	0.2	1.3
	Cyromazine	0.125–2	0.8 (0.5–1.7)	2.2 ± 0.3	4.2 (3)	0.2	1.9
	Pyriproxyfen	0.016–0.25	0.1 (0.03–0.1)	1.6 ± 0.3	2.9 (3)	0.4	5.0
	Methoxyfenozide	4–64	11.6 (5.8–19.7)	2.2 ± 0.3	4.5 (3)	0.2	1.2

Excluding the control (30 adults or larvae), 150 adults or larvae were tested in each bioassay. Conc. = Concentration. ^†^ Median lethal concentration, FL = fiducial limit. SE = Standard error. ^‡^ Performance ratio (was calculated as LC_50_ of insecticide in Alpha-Sel/LC_50_ of insecticide in Alpha-Unsel).

## Data Availability

The data presented in this study are available from the corresponding author on reasonable request.

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
