# Peer review of "Alpha-Cypermethrin Resistance in Musca domestica: Resistance Instability, Realized Heritability, Risk Assessment, and Insecticide Cross-Resistance"

_insects, 2023, doi:10.3390/insects14030233_

Round 1
Reviewer 1 Report
This research targets at Musca domestica L., which is a major carrier of serious diseases in humans and livestock. Due to the overuse of insecticides on pests, environmental pollution, endager public health, and especially enhance insecticide resistance in the target insect vectors. This paper intended to screen out different insecticides (absent or low CR associated with alpha-cypermethrin resistance) and Alpha-cypermethrin to do rotation application in M. domestica. A lot of bioassay work has been done in this paper, and the methods are introduced in detail, but still have some problems to be confirmed and details need to be corrected.
Abstract:
1. Line 11 & 24: Change “humans and livestock” to “humans and livestocks”.
2. Line 26: alpha-cypermethrin resistance should add the abbreviation (DR), corresponding to the text.
Introduction:
3. Line 46: The format of citations should be uniform.
4. Line 57: “Musca domestica” change to “M. domestica”, the same correction should be checked all the manuscript.
5. Line 62-63: In introduction, each sentence should have reference to support.
Materials and methods:
6. Line 119-121: “the alpha-cypermethrin–susceptible strain (Alpha-Sus) was maintained for 24 generations and the alpha-cypermethrin–selected strain (Alpha-Sel) was screened continuously with different concentrations of alpha-cypermethrin for 23 generations”, which means that Alpha-Sus and Alpha-Sel breeding are off by a generation. But judging by the results (table 3, table 4 and table 6), it was both G24, which is a little confusing.
7. Line 146: How much insect growth regulators to add to 500 ml of larval food.
8. Line 147: Is this supposed to be “150 larvae”, instead of “250 larvae per replicate” ?
9. Line 153: Latin should be abbreviated. This need to be checked throughout the manuscript.
10. Line 175: Change “p″ ”to “p”.
Results:
11. Line 200: Latin need italics.
12. Line 202: Note the LC50 format, “LC50”.
13. Line 205: Change “Com-pared” to “Compared”.
14. Line 208: What are the three alpha-cypermethrin–selected strain concentrations in Table 2 set according to? It was confused that G1-G8, G9-G17, G18-G23 are the same concentration, respectively. What is the connection between the concentration setting and the generation. This should be described in the paper.
15. Line 209: Table 3 Alpha-Sel has G24, but there are only G23 in table2 and Line121. Please check it.
16. Line 224: Table 4 mentioned Alpha-Unsel, which means without exposure to alpha-cypermethrin? Alpha-Unsel need to describe in methods 2.3 together with Alpha-Sus and Alpha-Sel, this will make it clear to the reader.
17. Line 246: Fig. 1A need to refine. Some information is blocked, eg. blue line of h2=0.18 on the top right corner and the X-axis coordinates.
18. Line 259: Fig. 2 need to refine. Some information is blocked, 2A, the X-axis coordinates; 2B, blue line of slope=2.1 on the top right corner and the green curve of slope=4.1.
Discussion:
19. The discussion section did not discuss important issues and repeated many descriptions of results. Although many references were cited, they were all superficial. Resistance instability, realized heritability, risk assessment, and insecticide cross-resistance. Through the results of these four aspects, it can refer to other articles respectively to provide constructive suggestions for production practice for insecticide rotation in M. domestica.
Conclusion:
20. The insecticides name of no or low CR associated with alpha-cypermethrin resistance should be summarized here.
Author Response
This research targets at Musca domestica L., which is a major carrier of serious diseases in humans and livestock. Due to the overuse of insecticides on pests, environmental pollution, endager public health, and especially enhance insecticide resistance in the target insect vectors. This paper intended to screen out different insecticides (absent or low CR associated with alpha-cypermethrin resistance) and Alpha-cypermethrin to do rotation application in M. domestica. A lot of bioassay work has been done in this paper, and the methods are introduced in detail, but still have some problems to be confirmed and details need to be corrected.
Thanks for the valuable comments and suggestions to improve the manuscript. We carefully responded to all comments.
Abstract:
- Line 11 & 24: Change “humans and livestock” to “humans and livestocks”.
Response: Done as suggested
- Line 26: alpha-cypermethrin resistance should add the abbreviation (DR), corresponding to the text.
Response: DR is used for instability, we wrote with the instability of the resistance trait.
Introduction:
- Line 46: The format of citations should be uniform.
Response: Done
- Line 57: “Musca domestica” change to “M. domestica”, the same correction should be checked all the manuscript.
Response: Done throughout the manuscript
- Line 62-63: In introduction, each sentence should have reference to support.
Response: Done as suggested
Materials and methods:
- Line 119-121: “the alpha-cypermethrin–susceptible strain (Alpha-Sus) was maintained for 24 generations and the alpha-cypermethrin–selected strain (Alpha-Sel) was screened continuously with different concentrations of alpha-cypermethrin for 23 generations”, which means that Alpha-Sus and Alpha-Sel breeding are off by a generation. But judging by the results (table 3, table 4 and table 6), it was both G24, which is a little confusing.
Response: Corrected
- Line 146: How much insect growth regulators to add to 500 ml of larval food.
Response: 140 ml of insect growth regulator solution was mixed with the larval food consisted of wheat bran (40.0 g), yeast (10.0 g), sugar (3.0 g), and dry milk powder (3.0 g).
- Line 147: Is this supposed to be “150 larvae”, instead of “250 larvae per replicate” ?
Response: Corrected
- Line 153: Latin should be abbreviated. This need to be checked throughout the manuscript.
Response: Done
- Line 175: Change “p″ ”to “p”.
Response: Done
Results:
- Line 200: Latin need italics.
Response: Done
- Line 202: Note the LC50 format, “LC50”.
Response: Done
- Line 205: Change “Com-pared” to “Compared”.
Response: Done
- Line 208: What are the three alpha-cypermethrin–selected strain concentrations in Table 2 set according to? It was confused that G1-G8, G9-G17, G18-G23are the same concentration, respectively. What is the connection between the concentration setting and the generation. This should be described in the paper.
Response: Described in the M&M section 2.3.
- Line 209: Table 3 Alpha-Sel has G24, but there are only G23in table2 and Line121. Please check it.
Response: Corrected
- Line 224: Table 4 mentioned Alpha-Unsel, which means without exposure to alpha-cypermethrin? Alpha-Unsel need to describe in methods 2.3 together with Alpha-Sus and Alpha-Sel, this will make it clear to the reader.
Response: Done
- Line 246: Fig. 1A need to refine. Some information is blocked, eg. blue line of h2=0.18 on the top right corner and the X-axis coordinates.
Response: Corrected
- Line 259: Fig. 2 need to refine. Some information is blocked, 2A, the X-axis coordinates; 2B, blue line of slope=2.1 on the top right corner and the green curve of slope=4.1.
Response: Corrected
Discussion:
- The discussion section did not discuss important issues and repeated many descriptions of results. Although many references were cited, they were all superficial. Resistance instability, realized heritability, risk assessment, and insecticide cross-resistance. Through the results of these four aspects, it can refer to other articles respectively to provide constructive suggestions for production practice for insecticide rotation in M. domestica.
Response: Unnecessary references are deleted and discussion is improved
Conclusion:
- The insecticides name of no or low CR associated with alpha-cypermethrin resistance should be summarized here.
Response: Done
Reviewer 2 Report
This study investigated the house fly insecticide resistance after selection using alpha-cypermethrin. Overall, experiment design is fine and results provide new information about the speed of resistance development and stability of the resistance trait. There are several major issues that need authors to justify.
How the author decided the concentration used in population selection? Why the author decides 12 generation as a phase to calculate heritability?
The author evaluated the performance of commercial insecticides against the selected population, but some values of LC were not reliable (based on vary large value of FL). This may be due to the author chose commercial products as target to evaluate the performance. Plus, the authors used the term “cross-resistance”. However, this study used commercial products on selected population (G24) to test cross resistance. This is not common in insecticide resistance studies. Authors need to use technical grade chemical to document the exact resistance levels. This is because different products contain different concentrations of a.i. The undisclosed inert ingredients of the products may also affect insect mortality. Authors do not know what caused mortality and cannot estimate the resistance levels to the active ingredient in the products. The different mortalities from different insecticide products are better to be described using performance ratios (PRs) or other terms rather than resistance ratios (RRs).
The authors calculated heritability in this study. But did not discuss the implications of heritability in the resistance study. There are also many minor format problems in the content. I have added comments directly to the manuscript. Here are some additional comments:
Line 118: why G24 of unselected population turned to be Alpha-sus? The author did not mention or define in anywhere.
Line 183: why 12 generation was defined as a phase?
Line 202: LC50 should be LC50
Line 205: Com-pared, there should not be a “-“.
Line 205: why use G24 of unselected population as the reference to calculate RR for all selected generations? Why not use the corresponding generation, ei, G10 / G10 to calculate RR in each generation?
Line 208: Suggest delete the number of survivals of male and female in the table (Table 2). The title of each column did not separate well. Please re-format.
Table 3. Where are the results of G2 to G4?
Line 183: How authors defined 12 generation as a phase? There is not definition in material and methods paragraph.
Table 5: Please re-format the table. For instance, “Female flies” were in two lines and “Male flies” was in one line. The annotation of mean surviving only mentioned female but not male.
Table 6: The fiducial limits of bifenthrin, deltamethrin, and cyfluthrin were too large. The large FL suggested the analysis of LC50 is not reliable. In addition, the RR used in in this bioassay was suggested change to PR (performance ratio). PR fits more with authors’ purpose.
Line 267: 15.47-fold in the content but 15.5 in Table 6. So as other values. Please only keep one digit after the decimal point.

Author Response
Reviewer 2
This study investigated the house fly insecticide resistance after selection using alpha-cypermethrin. Overall, experiment design is fine and results provide new information about the speed of resistance development and stability of the resistance trait. There are several major issues that need authors to justify.
Thanks for the valuable comments and suggestions to improve the manuscript. We carefully responded to all comments.
How the author decided the concentration used in population selection? Why the author decides 12 generation as a phase to calculate heritability?
Response: The concentrations were decided on the basis of the availability of a sufficient number of insects to survive for the next progeny. Half of the total generations were considered for each phase. There were 24 generations, so half will be 12 generations. Therefore 12 generations in each phase were used. A lot of literature is available to use half of the generation for each phase (e.g. Abbas et al. 2015, Shah et al. 2015a,b, Abbas et al. 2016, Afzal et al 2020)
The author evaluated the performance of commercial insecticides against the selected population, but some values of LC were not reliable (based on vary large value of FL). This may be due to the author chose commercial products as target to evaluate the performance. Plus, the authors used the term “cross-resistance”. However, this study used commercial products on selected population (G24) to test cross resistance. This is not common in insecticide resistance studies. Authors need to use technical grade chemical to document the exact resistance levels. This is because different products contain different concentrations of a.i. The undisclosed inert ingredients of the products may also affect insect mortality. Authors do not know what caused mortality and cannot estimate the resistance levels to the active ingredient in the products. The different mortalities from different insecticide products are better to be described using performance ratios (PRs) or other terms rather than resistance ratios (RRs).
Response: Changed to PRs
The authors calculated heritability in this study. But did not discuss the implications of heritability in the resistance study. There are also many minor format problems in the content. I have added comments directly to the manuscript. Here are some additional comments:
Response: Provided heritability implications in the discussion. All comments on the PDF are also carefully responded to.
Line 118: why G24 of unselected population turned to be Alpha-sus? The author did not mention or define in anywhere.
Response: Changed to Alpha-Unsel
Line 183: why 12 generation was defined as a phase?
Response: Half of the total generations were considered for each phase. There were 24 generations, so half will be 12 generations. Therefore 12 generations in each phase were used. A lot of literature is available to use half of the generation for each phase (e.g. Abbas et al. 2015, Shah et al. 2015a,b, Abbas et al. 2016, Afzal et al 2020)
Line 202: LC50 should be LC50
Response: Done
Line 205: Com-pared, there should not be a “-“.
Response: Corrected
Line 205: why use G24 of unselected population as the reference to calculate RR for all selected generations? Why not use the corresponding generation, ei, G10 / G10 to calculate RR in each generation?
Response: Because Alpha-Unsel (G24) shows the lowest LC50, therefore G24 was used as a susceptible reference strain for comparison. We calculated RRs by using each respective generation as suggested and shown in Fig. 1
Line 208: Suggest delete the number of survivals of male and female in the table (Table 2). The title of each column did not separate well. Please re-format.
Response: Deleted number of survivals and formatted table
Table 3. Where are the results of G2 to G4?
Response: Unfortunately, Bioassay were not performed on these generations
Line 183: How authors defined 12 generation as a phase? There is not definition in material and methods paragraph.
Response: Response: Half of the total generations were considered for each phase. There were 24 generations, so half will be 12 generations. Therefore 12 generations in each phase were used. A lot of literature is available to use half of the generation for each phase (e.g. Abbas et al. 2015, Shah et al. 2015a,b, Abbas et al. 2016, Afzal et al 2020). Moreover described in M & M section
Table 5: Please re-format the table. For instance, “Female flies” were in two lines and “Male flies” was in one line. The annotation of mean surviving only mentioned female but not male.
Response: Rephrased and corrected
Table 6: The fiducial limits of bifenthrin, deltamethrin, and cyfluthrin were too large. The large FL suggested the analysis of LC50 is not reliable. In addition, the RR used in in this bioassay was suggested change to PR (performance ratio). PR fits more with authors’ purpose.
Response: Changed to PR
Line 267: 15.47-fold in the content but 15.5 in Table 6. So as other values. Please only keep one digit after the decimal point.
Response: Done
Round 2
Reviewer 1 Report
This modified version is much better than the original version. All the comments have been solved. So I agree to accept this paper.
Author Response
Thanks for accepting the manuscript.
Reviewer 2 Report
The revised version looks good. I only suggest a few minor changes.

Author Response
Thanks for the minor corrections. All the suggestions and comments are responded to in the manuscript through track changes.
Thanks
